# Influence of Biologically Oriented Preparation Technique on Peri-Implant Tissues; Prospective Randomized Clinical Trial with Three-Year Follow-Up. Part II: Soft Tissues

**DOI:** 10.3390/jcm8122223

**Published:** 2019-12-16

**Authors:** Rubén Agustín-Panadero, Naia Bustamante-Hernández, Carlos Labaig-Rueda, Antonio Fons-Font, Lucía Fernández-Estevan, María Fernanda Solá-Ruíz

**Affiliations:** Department of Dental Medicine, Faculty of Medicine and Dentistry, University of Valencia, 46010 Valencia, Spain; rubenagustinpanadero@gmail.com (R.A.-P.); carlos.labaig@uv.es (C.L.-R.); antonio.fons@uv.es (A.F.-F.); lucia.fernandez-estevan@uv.es (L.F.-E.); m.fernanda.sola@uv.es (M.F.S.-R.)

**Keywords:** implant-supported prosthesis, complications, soft tissues, bone loss and clinical parameters, screw-retained, cemented, BOPT

## Abstract

***Purpose***: The objective of this prospective randomized clinical trial (RCT) was to analyze and compare the clinical behavior of three types of prosthesis supported by single implants in the posterior region after three years of functional loading. ***Materials and methods***: Seventy-five implants were divided into three groups according to the type of prosthetic restoration: screw-retained crown (group GS); cemented crown without finishing line (biologically oriented preparation technique) (group GBOPT); and conventional cemented crown with finishing line (group GCC). After three years in function, clinical parameters (presence of keratinized mucosa, probing depths, bleeding on probing, and radiographic bone loss) were compared between the three experimental groups. The possible correlation between soft tissue clinical parameters and bone loss was also analyzed. ***Results***: Statistical analysis found significant differences in clinical parameters between the different types of crown, with the cemented restoration without finishing line (BOPT) presenting fewer complications and better peri-implant health outcomes including: significantly different KMW data (mm), with significant differences between groups GBOPT and GCC (*p* < 0.001, Kruskal–Wallis test), with GBOPT obtaining larger quantities of keratinized mucosa (KM); statistically significant differences in probing depth (PD) values between groups GBOPT and GCC (*p* = 0.010, Kruskal–Wallis test); significant differences in bleeding on probing (BOP) between groups GBOPT and GCC (*p* = 0.018, Chi^2^ test) in favor of GBOPT. ***Conclusions***: Soft tissue behavior around implants is related to the type of prosthetic restoration used, with cemented prostheses with BOPT presenting better peri-implant soft tissue behavior.

## 1. Introduction

There is a direct relationship between the health of peri-implant mucosa and the health of the underlying bone [1,2]. Maintaining the health and stability of peri-implant soft tissues, as well their successful integration, is a key clinical parameter for the long-term success of implant-supported prosthetic restorations [3,4,5,6]. After prosthetic loading, peri-implant soft tissue stability would appear to correlate to hard tissue changes over time, as it prevents the apical migration of the epithelium, which in turn dictates the quantity of subsequent bone resorption [6,7,8,9]. Biologically oriented preparation technique (BOPT) is a type of preparation without finish line. This protocol eliminates the crown’s anatomic emergence profile, creating a new anatomic crown with a prosthetic emergence profile that simulates the shape of a natural tooth [1].

Long-term implant success is partly determined by the soft tissue collar established around the implant restoration and implant neck, as this can provide an effective seal that protects against bacterial invasion and possible future inflammation [10]. This type of restoration without finishing line can also be used for implant-supported full coverage crowns, also known as the biologically oriented preparation technique (BOPT). The design improves peri-implant tissue behavior as it eliminates the gap between the restoration and the finishing line at the end of the transepithelial abutment. This leaves the apical portion of the abutment free of coverage by the prosthetic restoration for at least 2 mm in order to stabilize the adjacent connective tissue. It has been suggested that a minimum peri-implant mucosa thickness is needed to protect ossteointegration after placing the abutment and loading the implant [9,11].

In addition to radiography, there are various tools and indicators that help to assess the clinical behavior of peri-implant tissues: implant mobility, probing depth (PD), gingival and plaque indices, bleeding on probing (BOP), and the width of keratinized mucosa (KMW) [12,13,14]. The literature expresses a great deal of controversy regarding the role played by keratinized mucosa (KM) surrounding implants as a barrier against microorganisms and subgingival plaque [15,16]. Nevertheless, a width of 2 mm of KM, including the gingival margin and the mucogingival junction [14] with at least 1 mm of attached mucosa has been proposed as adequate to ensure gingival health [17].

Araujo and Lindhe [15,16] concluded that peri-implant health means an absence of clinical signs of inflammation (erythema and swelling) and an absence of BOP (Figure 1).

In healthy tissue, when PD is measured, probe penetration is deeper around implants than around natural teeth [16]. Long-term clinical studies have shown that PD in healthy peri-implant mucosa may exceed 4 mm but is not usually deeper than 6 mm [16,17,18,19]. Absence of BOP is a good predictor of tissues’ future stability [17]. Other possible complications that may affect implant-supported restorations are either mechanical or biological. While cemented restorations may present more severe biological complications (peri-implant inflammation) as a consequence of remnants of excess cement [20,21], screw-retained restorations suffer less serious problems, as these are likely to be technical/mechanical such as screw loosening/fracture or chipping [22,23,24,25].

The aim of this randomized clinical trial (RCT) was to analyze the clinical behavior of peri-implant soft tissues around three types of implant-supported prosthetic restorations (screw-retained, cemented BOPT, and conventional cemented with finishing line) after three years of functional loading. In addition, the clinical behavior of screw-retained crowns, crowns cemented slightly subgingivally without finishing line (BOPT), and crowns cemented conventionally with finishing line placed slightly subgingivally were compared. A comparative analysis was also performed between screwed and cemented prostheses (without distinguishing between BOPT and conventional cementation). The mean KMW, PD, BOP, and complications between the two cementation techniques (BOPT/conventional) were also analyzed. Differences between the maxilla and mandible were analyzed for each type of prosthesis. The hypotheses tested were that there would be: (1) better clinical peri-implant soft tissue behavior with cemented crowns compared with screw-retained restorations; (2) similar KMW, PD, BOP, and complications between the two types of cemented crown (BOPT/conventional); and (3) worse clinical peri-implant soft tissue behavior with the implants placed in the maxilla than in the mandible.

## 2. Materials and Methods

This prospective RCT was conducted at the dental clinic of the Prosthodontic and Occlusion Unit, Faculty of Dentistry and Medicine, University of Valencia (Carrer Gasco Oliag Nº1, 46010, Valencia, Spain). The trial was approved by the University of Valencia Ethics Committee for Research Involving Humans (registration number 111508667100076). All participants were provided with full information about the trial, its objectives, and the procedures involved, and all gave their informed consent to take part before the trial began.

The study sample consisted of 75 implant-supported crowns. Patients were treated between January 9th and December 16th 2015. Implant surgery and prosthetic restoration placement were performed during this period, with the follow-up period commencing in January 2016 and ending in January 2019. Inclusion and exclusion criteria have been described in Table 1. The implants used in the trial were Vega Klockner implants (Escaldes-Engordany, Andorra) with internal hexagonal connection associated with conical geometry. A standardized protocol for implant placement was followed for all cases. The main inclusion criteria were that only patients in need of single implants were included who had good periodontal health and at least 2 mm of vestibular keratinized mucosa prior to treatment. Moreover, all patients presented conditions that made it possible to place the implant without bone regeneration, to raise flaps without release incisions, and without detaching the mucogingival line in order to avoid modifying the position of the buccal mucosa and therefore its keratinized mucosa. All implants were placed in premolar-molar posterior regions (Table 1).

After surgery, patients returned for check-ups one week, one month, and three months later, when prosthetic loading was performed.

Before placing the prosthetic restorations, the sample was randomized into three groups according to the prosthetic type used to restore the implants, using online randomization software www.alazarinfo.es: group GS, implants restored with CAD/CAM (Computer-aided design/computer-aided manufacturing) crowns (Cr-Co with feldspathic ceramic coating) screwed directly onto the implants; group GBOPT, implants restored with crowns cemented slightly subgingivally (0.5 mm) onto grade IV titanium abutments without finishing line (BOPT); group GCC, implants restored with crowns cemented conventionally onto grade IV titanium abutments with 1 mm wide chamfered finishing line placed slightly subgingivally (0.5 mm).

All implants were restored with metal-ceramic crowns milled from Cr-Co (Archimedes, Klockner S.A. Barcelona, Spain) and coated with feldspathic ceramic (IPS d-Sign, Ivoclar Vivadent, Schaan, Liechtenstein) fabricated by means of CAD/CAM design software (EXOCAD DENTAL CAD, Exocad America, Inc., STI Holdings, Woburn, MA, USA) (Figure 2).

All implants restored with cemented crowns (BOPT and conventional cemented) were cemented using provisional cement (Premier Implant Cement, Premier Dental, Plymouth Meeting, PA, USA). Excess cement was cleaned as follows: interproximal dental floss was placed before the cement set; cleaning was carried out with silk, moving it towards the implant’s prosthetic platform, taking advantage of the fact that the cement is radiopaque so excess cement can be seen in radiographic control; then, excess cement was removed with a fine instrument.

The present RCT analyzed the clinical behavior of peri-implant soft tissues, evaluating the quantity of KM (mm), PD (mm), and BOP, as well as any mechanical or biological complications. Thereafter, correlations between bone loss and soft tissue clinical variables were analyzed. Soft tissue variables were recorded at the moment of prosthetic loading and after 3 years.

### 2.1. Clinical Evaluation of Peri-Implant Soft Tissues

To evaluate changes to peri-implant soft tissues, periodontal examination was conducted by one of the authors (A.-P.,R.) using a periodontal probe (N116, Offset, Double End, Color Coded, with Markings, Nordent Mfg Inc., Elk Grove Village, IL, USA). The clinical parameters recorded included the width of keratinized mucosa (KMW) measured in the vestibular area of peri-implant mucosa. KMW was measured from the crown’s mucosa margin to the mucogingival junction, placing the probe parallel to the longitudinal axis of the implant as proposed in a study by Schwarz et al. [26]. The protocol was described as: the distance between the gingival margin and the mucogingival junction at the mid-buccal aspect of the implant measured in millimeters. The crown was taken as a fixed immoveable reference for taking measurements (Figure 3).

PD and BOP were recorded, as described by Ainamo and Bay [27]. These evaluations were performed at the moment of crown placement and after 3 years of functional loading (Figure 4).

Possible biological or mechanical complications were analyzed. At each evaluation time the state of the screw and abutments (presence of loosening or fracture), and the integrity of the ceramic coating (presence of fracture or chipping) were analyzed, as well as biological complications such as peri-implant mucositis and/or peri-implantitis.

For the accuracy and good reproducibility of measurements, a calibration exercise was performed until the coefficient agreement reached 90%. The calibration was achieved by assessing three patients twice in a period of 24 h. Calibration was accepted if the variability between the repeated measurements at baseline and at 24 h was under 3%.

### 2.2. Statistical Analysis

Descriptive statistics were calculated for all clinical variables (mean, standard deviation, range and median) applying a non-parametric approach (Chi^2^ test and Fisher exact test). The Kruskal–Wallis test was applied to assess the homogeneity of the variables KMW and PD according to group. The Mann–Whitney test was used in similar conditions for pairs of independent groups. Spearman’s correlation coefficient was used to measure the degree of non-linear correlation between continuous variables (such as KM loss). A power of 70.5% was provided to detect a medium-to-large effect size (d = 0.65). The significance level set in all analyses was 5%.

## 3. Results

### Study Population

This prospective RCT included 75 implants placed in 75 patients (49 women and 26 men, with a mean age of 42.7 ± 10.6 years) (Table 2).

Patients/implants were divided into three groups (n = 25) according to the type of prosthetic restoration used. Peri-implant soft tissue evaluation was carried out for each group after three years of functional loading. It should be noted that seven patients (four in GBOPT and three in GCC) were lost to the trial due to their failure to attend follow-up visits, leaving a sample of 68 implants in 68 patients.

Of the total implant sample, 42.6% were placed in the maxilla and 57.4% in the mandible; 55.9% of the implants were placed in molar and 44.1% in premolar regions.

Part I of the present RCT analyzed radiographic peri-implant bone, while the second part analyzed the following soft tissue variables: KMW (mm), PD (mm), presence of BOP (no/yes), and possible biological and/or mechanical complications.

Mean KMW around screw-retained prostheses (group GS) was 2.00 ± 0.98 mm. But around cemented BOPT prostheses (group GBOPT) it was 2.74 ± 0.72 mm, while conventional cemented prostheses (group GCC) obtained a mean KMW of 1.23 ± 0.55 mm. The three groups showed significantly different KMW data (mm), with significant differences between groups GBOPT and GCC (*p* < 0.001, Kruskal–Wallis test), with GBOPT obtaining greater quantities of KM. group GS obtained higher KMW than group GCC (*p* = 0.005, Kruskal–Wallis test), and group GBOPT obtained better KMW results than group GS (*p* = 0.011, Kruskal–Wallis test) (Figure 5).

Analyzing these results in relation to implant position (maxilla/mandible), differences were clear between the two types of cemented prostheses (groups GBOPT and GCC) (*p* = 0.001, Kruskal–Wallis test); although differences did not reach statistical significance between the other groups, group GS and group GCC (*p* = 0.299, Kruskal–Wallis test), or between group GS and group GBOPT (*p* = 0.051, Kruskal–Wallis test). In the mandible, differences in KMW between groups were evident between groups GBOPT and GCC (*p* < 0.001, Kruskal–Wallis test) and between group GS and group GCC (*p* = 0.006, Kruskal–Wallis test), but no significant difference was found between groups GS and GBOPT (*p* = 0.238, Kruskal–Wallis test).

Comparing the influence of prosthesis type between screw-retained and cemented prostheses (combining the two cemented groups GBOPT + GCC), KMW values were similar (*p* = 0.847, Mann–Whitney test) (Figure 6).

No significant differences between screw-retained and cemented prostheses were found when these were positioned in the maxilla, but in the mandible cemented prostheses showed significantly less KMW (*p* = 0.011, Mann–Whitney test).

Regarding the PD variable, mean PD in the screw-retained group (group GS) was 2.00 ± 1.19 mm, while in the cemented BOPT group (group GBOPT) mean PD was 1.90 ± 0.94 mm; in the conventional cemented prostheses group (group GCC) it was 3.05 ± 1.53 mm. Statistically significant differences between PD values were found, with significant differences between groups GBOPT and GCC (*p* = 0.010, Kruskal–Wallis test), and between groups GS and GCC (*p* = 0.007, Kruskal–Wallis test), although values were similar between groups GS and GBOPT (*p* = 1.000, Kruskal–Wallis test) (Figure 7).

Analyzing the influence of implant position (mandible/maxilla) on PD, differences were identified in the mandible, although only between the GS and GBOPT groups (*p* = 0.021, Kruskal–Wallis test). Analyzing differences in PD between screw-retained and cemented prostheses (combining the two cemented groups GBOPT + GCC), no statistically significant differences were found (*p* = 0.108, Mann–Whitney test) (Figure 8). This was due to the fact that in the cemented group the values were more disperse as merging the two groups (GBOPT + GCC) doubled the sample size compared with screw-retained crowns (GS). Values differed widely among cemented prostheses (lower PD values in GBOPT but higher PD values in GCC), causing data dispersion as shown in the box plot (Figure 8).

Nor were significant differences observed in relation to implant position (mandible/maxilla), although a strong tendency was found towards greater PD in the mandible in the group of conventional cemented prostheses (GCC) (*p* = 0.084, Mann–Whitney test).

The third variable analyzed was the presence of BOP. In group GS (screw-retained), 24.0% of the sample presented BOP. In group GBOPT (cemented BOPT prostheses), 9.5% of the sample presented BOP, while in group GCC (conventional cemented prostheses) 40.9% of cases presented BOP. Differences in BOP rates were situated on the threshold of significance (*p* = 0.059). Differences were clearly significant between groups GBOPT and GCC (*p* = 0.018, Chi^2^ test) but not between groups GS and GCC (*p* = 0.215, Chi^2^ test), or between groups GS and GBOPT (*p* = 0.260, Chi^2^ test).

The possible influence of implant placement in the mandible or maxilla on BOP did not show any significant differences in any of the groups, nor were significant differences obtained between screw retained prostheses (GS) and cemented prostheses (GBOPT + GCC) (*p* = 0.885, Chi ^2^ test) (Figure 9).

Lastly, any mechanical or biological complications (bleeding on probing, recession, mucositis, and peri-implantitis, among others) occurring during the 3-year trial period were recorded (Table 3).

Only ten complications occurred during the 3-year trial: four cases of screw loosening, two of chipping, two decementations, and two biological complications (peri-implantitis) (Figure 10).

Differences in rates of implants suffering complications between groups were on the threshold of statistical significance (*p* = 0.062, Chi ^2^ test). No differences were found between groups GBOPT and GCC (*p* = 0.108, Chi ^2^ test) or between groups GS and GCC (*p* = 0.627, Chi ^2^ test). However, the rate of complication in the GS group was significantly higher than in the BOPT group (*p* = 0.025, Chi^2^ test). Complication rates were not found to be influenced by implant position (mandible/maxilla). As no complications were recorded in the BOPT group, it was not possible to apply statistical tests. No significant differences were found between screw-retained (GS) and cemented (BOPT + GCC) prostheses, although a slight tendency was observed towards a higher rate of complications among screw-retained prostheses (*p* = 0.09, Chi ^2^ test).

All the variables analyzed in the present trial were correlated to determine any significant relations between them.

Bone loss (mm) was analyzed in part I of this study [28]. The relation between clinical peri-implant soft tissue variables and bone loss is shown in the following table (Spearman’s coefficient and Mann–Whitney test) (Table 4).

A clear correlation was found between clinical peri-implant soft tissue parameters and bone loss. There was a strong negative relation between KMW and bone loss: the greater the KMW, the lower the extent of peri-implant bone loss (negative coefficient). This relation between KMW and bone loss was statistically significant for all three groups: screw-retained (group GS), cemented BOPT (group GBOPT) and conventional cemented prostheses (group GCC) (*p* < 0.001, Spearman and Mann–Whitney test) (Figure 11).

The correlation between PD and bone loss was also significant in all three groups, presenting the same significance level in groups GS and GCC (*p* < 0.001, Spearman and Mann–Whitney test), while for group GBOPT significance was *p* = 0.008 (Spearman and Mann–Whitney test): the greater the probing depth, the greater the peri-implant bone loss. The association between BOP and bone loss was also statistically significant, showing the same relation in groups GS and GCC (*p* < 0.001, Spearman and Mann–Whitney test), while the significance level was lower in group GBOPT (*p* = 0.019, Spearman and Mann–Whitney test) (Figure 12).

Lastly, it was found that bone loss measurements increased when BOP was present for all three types of prosthesis (Figure 13).

## 4. Discussion

This randomized clinical trial set out to assess the clinical behavior and possible complications of peri-implant soft tissues after three years of functional loading of three types of implant-supported prostheses.

KMW data (mm) showed significant differences between the three groups. Between groups GBOPT and GCC (*p* < 0.001, Kruskal–Wallis test), GBOPT obtained greater quantities of KM. group GS obtained higher KMW than group GCC (*p* = 0.005, Kruskal–Wallis test), and group GBOPT obtained better KMW results than group GS (*p* = 0.011, Kruskal–Wallis test).

Analyzing these results in relation to implant position (maxilla/mandible), differences were clear between the two types of cemented prostheses (groups GBOPT and GCC) (*p* = 0.001, Kruskal–Wallis test). In the mandible, differences in KMW between groups were evident between groups GBOPT and GCC (*p* < 0.001, Kruskal–Wallis test) and between group GS and group GCC (*p* = 0.006, Kruskal–Wallis test), whereby cemented prostheses (GCC) showed significantly less KMW (*p* = 0.011, Mann–Whitney test).

Regarding the PD variable, statistically significant differences were found between groups GBOPT and GCC (*p* = 0.010, Kruskal–Wallis test), and between groups GS and GCC (*p* = 0.007, Kruskal–Wallis test).

Analyzing the influence of implant position (mandible/maxilla) on PD, differences were identified in the mandible, although only between groups GS and GBOPT (*p* = 0.021, Kruskal–Wallis test).

The third variable analyzed was the presence of BOP. Differences were clearly significant between groups GBOPT and GCC (*p* = 0.018, Chi^2^ test), with GBOPT presenting less presence of BOP.

According to these results, the hypothesis that cemented crowns would present better clinical peri-implant soft tissue behavior than screw-retained restorations, and the hypothesis that cemented crowns (BOPT/conventional) would present similar KMW, PD, BOP values and complications, were rejected.

Lastly, any mechanical or biological complications (bleeding on probing, recession, mucositis, and peri-implantitis, among others) were recorded. The rate of complications in the GS group was significantly higher than in the BOPT group (*p* = 0.025, Chi^2^ test).

A clear correlation was found between clinical peri-implant soft tissue parameters and bone loss. There was a strong negative relation between KMW and bone loss: the greater the KMW, the lower the extent of peri-implant bone loss (negative coefficient). This relation between KMW and bone loss was statistically significant for all three groups (*p* < 0.001, Spearman and Mann–Whitney test).

The correlation between PD and bone loss was also significant in all three groups, presenting the same significance level in groups GS and GCC (*p* < 0.001, Spearman and Mann–Whitney test), while for the GBOPT group the significance was *p* = 0.008 (Spearman and Mann–Whitney test): the greater the probing depth, the greater the peri-implant bone loss. The association between BOP and bone loss was also statistically significant, showing the same relation in groups GS and GCC (*p* < 0.001, Spearman and Mann–Whitney test), while the significance level was lower in the GBOPT group (*p* = 0.019, *Spearman and Mann-Whitney test*) Lastly, it was found that bone loss measurements increased when BOP was present for all three types of restorations.

There was no difference in soft tissue behavior between implants placed in the maxilla or the mandible, so the hypothesis that implants placed in the maxilla would present worse clinical peri-implant soft tissue behavior than implants placed in the mandible was rejected.

Soft tissue evaluation was performed with a periodontal probe to analyze keratinized mucosa width (KMW) (mm), probing depth (PD) (mm), and the presence of bleeding on probing (BOP) (no/yes). Diagnosis of peri-implant health is usually performed by the methods established at the World Workshop 2017 as described by Renvert et al. [16], which are as follows: Visual inspection demonstrating the absence of peri-implant signs of inflammation: pink as opposed to red, no swelling as opposed to swollen tissues, firm as opposed to soft tissue consistency; Lack of profuse (line or drop) bleeding on probing; probing pocket depths could differ depending on the height of the soft tissue at the implant location. An increase in probing depth over time, however, conflicts with peri-implant health, and absence of further bone loss following initial healing, which should not be ≥2 mm. These parameters have been used in various trials to analyze peri-implant soft tissue behavior [29,30,31].

To date, no published research has analyzed implant-supported BOPT crowns compared with a control group or with conventional restorations over a horizontal finishing line, whether cemented or screw-retained. The few works published on implant-supported BOPT-type crowns and peri-implant soft tissue behavior have employed different methodologies making it impossible to carry out absolute comparisons with the present trial [29,30,32,33].

One of the limitations of the study is that the Pink Esthetic Score was not performed, which could have allowed a comparison with the data obtained by Canullo et al. [30], although this study did not include any control group, so comparison would only have been possible with the present study’s GBOPT group. It would be interesting to extend the follow-up period and analyze the evolution of the sample in years to come, and also to investigate the possible influence of abutment height and implant position in the dental arch (anterior vs. posterior).

Regarding the clinical soft tissue parameters analyzed in the present RCT, a clear correlation was observed between KMW and peri-implant bone loss, the latter being greater when KMW around the implants was smaller. Mobile mucosa facilitates biofilm penetration into the peri-implant sulcus, which triggers neutrophil and lymphocyte activation. For this reason, it has been argued that an adequate band of keratinized tissue around the implant is decisive in maintaining periodontal tissue stability [34]. Various articles have suggested that the accumulation of plaque and marginal inflammation are more frequent around implants with less than 2 mm of keratinized tissue [15]. Few studies have evaluated tissue responses around conical convergent abutments [29,32]. In the present trial, the GBOPT group (cemented crown without finishing line or BOPT), in addition to being the prosthesis that presented the least bone loss, also showed the greatest KMW, obtaining a mean of 2.74 ± 0.72 mm, achieving over 2 mm in all cases. It was noted that KMW loss was not very high since one of the inclusion criteria was that the patients had at least 2 mm of keratinized mucosa before treatment. Therefore, this always exceeded the inadequate KMW values of less than 2 mm proposed in the literature as a predisposing factor for plaque accumulation and peri-implant inflammation [13,14]. The fact that BOPT presented greater KMW could be due to abutment convergence which, as described above, increases the space available for connective tissue stabilization leading to increased KMW [30,32,33]. In the same way, the present trial observed a strong correlation between peri-implant bone loss and the clinical soft tissue parameters evaluated. Greater bone loss was accompanied by smaller KMW, greater PD, and higher probability of BOP. Not only could the use of conical abutments improve peri-implant bone levels, but it could also reduce the depth of the peri-implant sulcus. It has been suggested that the biological phenomenon of peri-implant bone preservation is related to the circular stabilization of connective tissue around the abutment and the presence of a superficial sulcus [29,35].

Regarding mechanical and biological complications, it should be noted that only ten complications occurred during the 3-year trial period, which were as follows: four cases of screw loosening, two cases of chipping, two decementations, and two biological complications (peri-implantitis). In particular, the group of cemented BOPT crowns presented no complications at all. Although the other groups of conventional cemented and screw-retained prostheses did present complications, these were not significant, although the rate of complication in the GS group (screw-retained) was significantly higher than the GBOPT group (cemented BOPT crowns) (*p* = 0.025). The GCC group (conventional cemented prostheses) suffered two biological complications (peri-implantitis) and two decementations. These findings concur with the literature, which reports that cemented restorations have been associated with higher rates of biological complications such as peri-implant inflammation, perhaps as a result of excess cement remnants [20,21,25]. Screw-retained prostheses are associated with higher rates of mechanical complications [22,23,24,25], as seen in the present trial with four cases of screw loosening and two of chipping in the GS group. Nevertheless, the advantages of screw-retained crowns should be noted, such as the reversibility of the treatment with a single component instead of two, which simplifies the restoration process; moreover, it is easy to resolve complications and these are generally less severe [36]. Other factors such as platform switching could influence the amount of bone loss. Platform-switching preserves more bone by placing the implant-abutment interface away from the crestal bone. As a consequence, the soft tissues attempt to sit on top of the dental implant creating a mechanical protective seal [29]. This factor has not been considered in the present investigation since all three groups presented platform switching, so that if this does constitute an influential factor, it would have exerted the same influence in the three groups. According to BOPT philosophy, restorations present double platform switching, with both a horizontal and a vertical change. It would appear that bone loss tends to decrease, as the restoration-abutment interface is moved away from the bone in two spatial planes.

This prospective RCT could help clinicians select prosthetic crown designs that achieve optimal clinical outcomes for peri-implant bone level preservation and soft tissue health and stability. The results show that implants with convergent abutments obtain the best outcomes in terms of bone loss and soft tissue parameters.

More RCTs are required to investigate BOPT in relation to the clinical behavior and complications associated with this type of implant-supported restoration. More research is also needed to determine the role played by different types of prosthesis in the development of complications and peri-implant pathologies.

## 5. Conclusions

On the basis of the results obtained in this RCT it may be concluded that:

The cemented BOPT prosthesis obtained greater KMW, less PD, and lower incidence of BOP after 3 years of functional loading in comparison with screw-retained and conventional cemented crowns, as well as remaining completely free of mechanical and biological complications.

A correlation exists between bone loss and peri-implant soft tissue parameters: the greater the KMW present, the less the peri-implant bone loss; the greater the PD, the greater the bone loss; and the greater the bone loss value the higher the numbers of cases presenting BOP, regardless of the type of prosthesis.

## Figures and Tables

**Figure 1 jcm-08-02223-f001:**
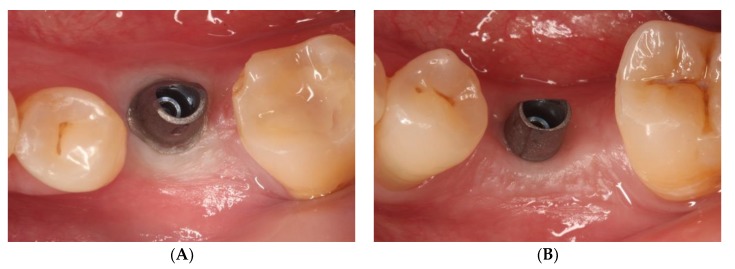
Moment of transepithelial abutment placement. (**A**) Abutment for conventional cemented prosthesis with horizontal finishing line; (**B**) Abutment for cemented prosthesis without finishing line or biologically oriented preparation technique (BOPT).

**Figure 2 jcm-08-02223-f002:**
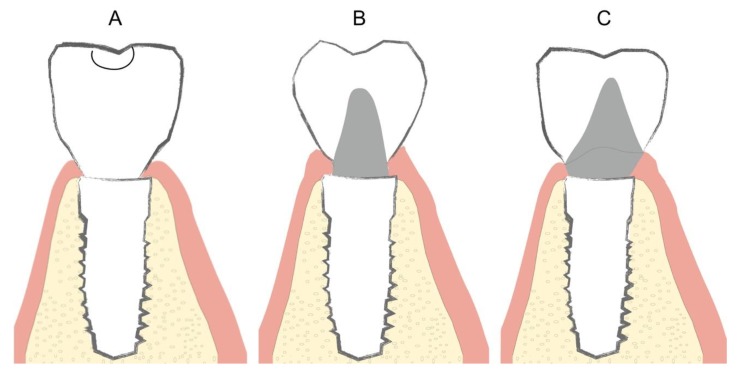
Three restoration types. (**A**): screw-retained crown (GS); (**B**): cemented without finishing line (GBOPT); (**C**): conventional cementation with finishing line (GCC).

**Figure 3 jcm-08-02223-f003:**
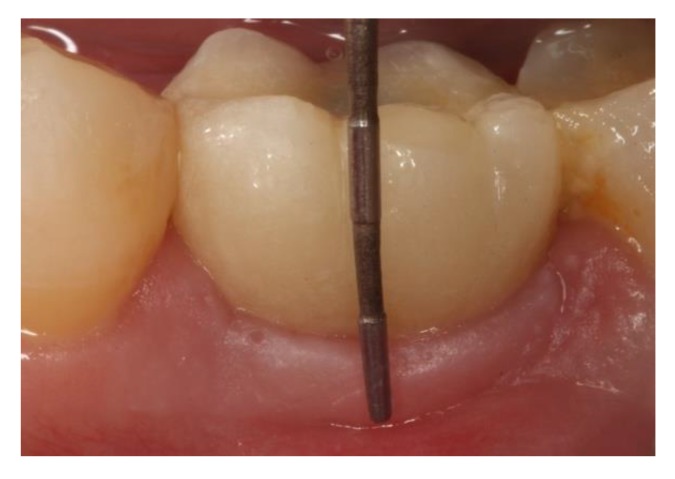
Width of keratinized mucosa (KMW) measurements, placing the probe parallel to the longitudinal axis of the implant.

**Figure 4 jcm-08-02223-f004:**
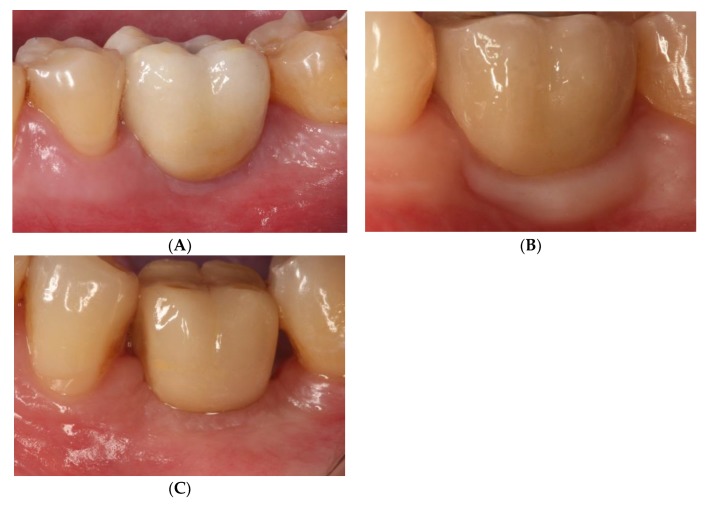
Healed mucosa without inflammation showing healthy peri-implant tissue. (**A**). Screw-retained crown (group GS). (**B**). Cemented BOPT crown (group GBOPT). (**C**). Conventional cemented crown (group GCC).

**Figure 5 jcm-08-02223-f005:**
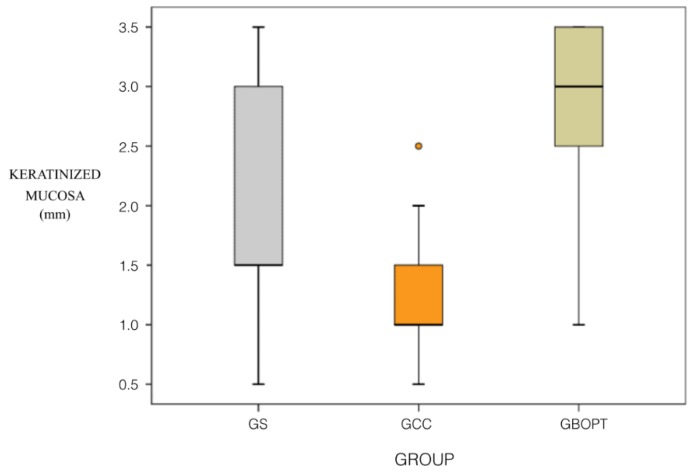
Box Plot: keratinized mucosa width (mm) in each group.

**Figure 6 jcm-08-02223-f006:**
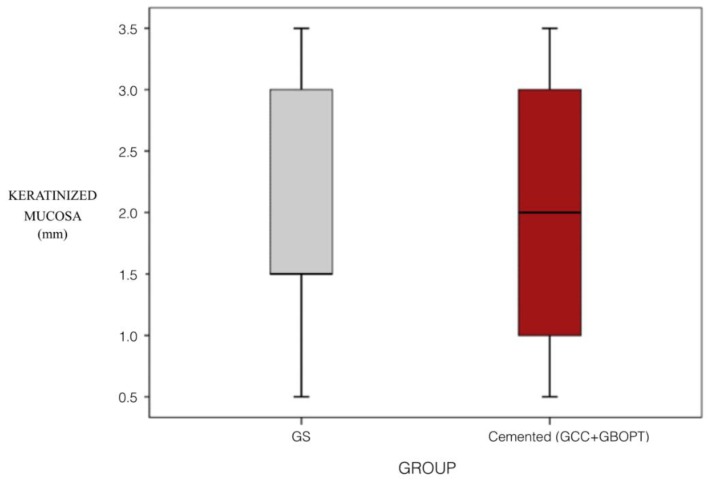
Box Plot: keratinized tissue width (mm) in relation to prosthetic type (screw-retained vs. cemented).

**Figure 7 jcm-08-02223-f007:**
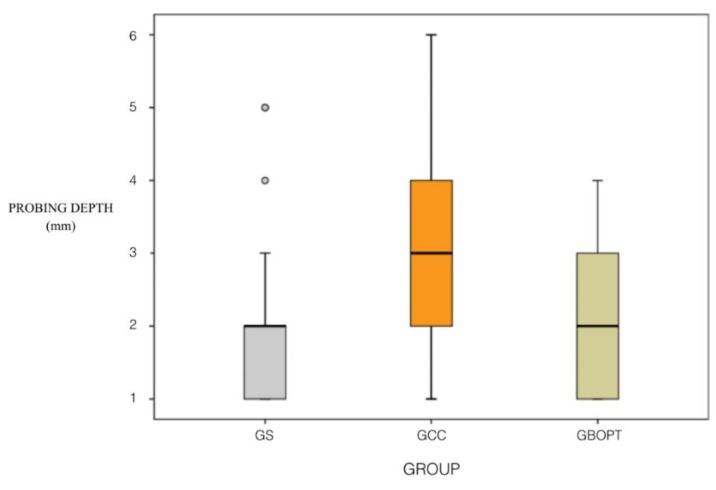
Box Plot: probing depth (mm) in each group.

**Figure 8 jcm-08-02223-f008:**
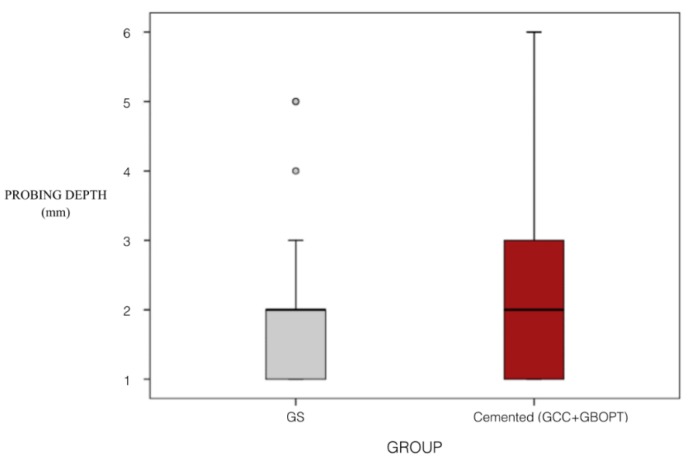
Box plot showing probing depth values (mm) in relation to prosthesis type (screw-retained vs. Cemented).

**Figure 9 jcm-08-02223-f009:**
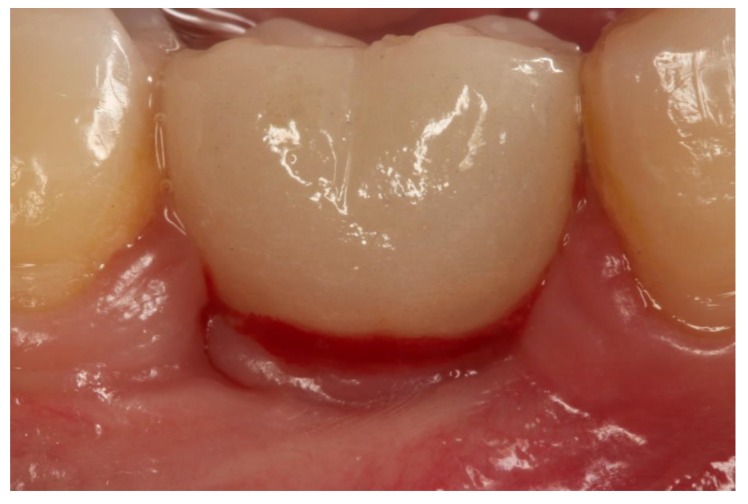
Case presenting bleeding on probing in a patient with at least 2 mm keratinized mucosa prior to treatment.

**Figure 10 jcm-08-02223-f010:**
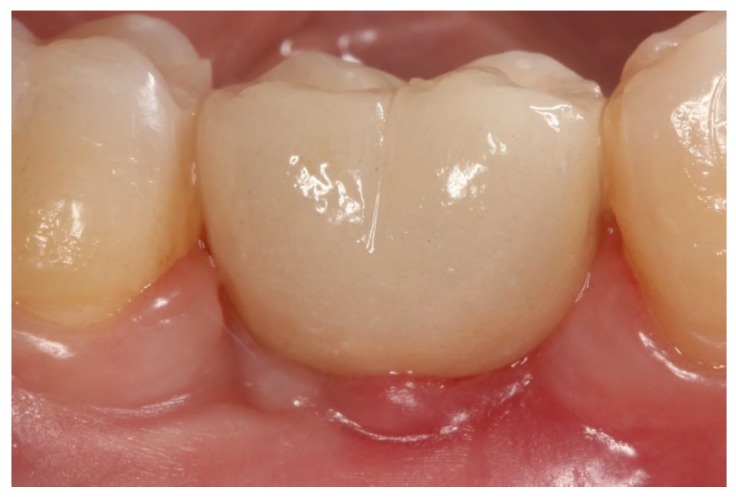
Restoration presenting inflammation in the surrounding peri-implant tissue.

**Figure 11 jcm-08-02223-f011:**
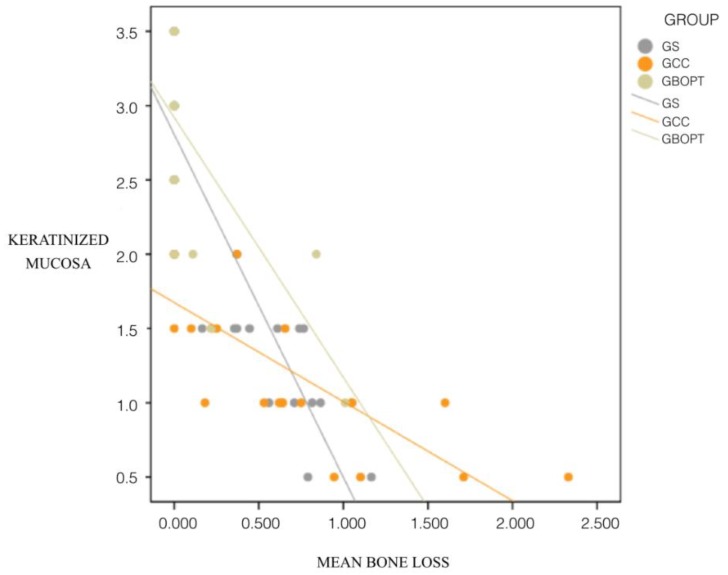
Correlation between keratinized mucosa width and peri-implant bone loss in each group.

**Figure 12 jcm-08-02223-f012:**
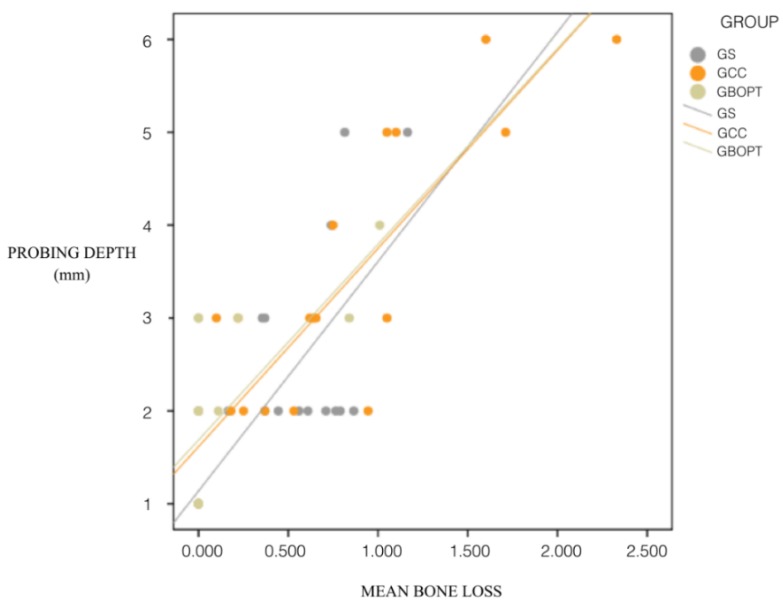
Correlation between keratinized mucosa width, probing depth, and peri-implant bone loss in each group.

**Figure 13 jcm-08-02223-f013:**
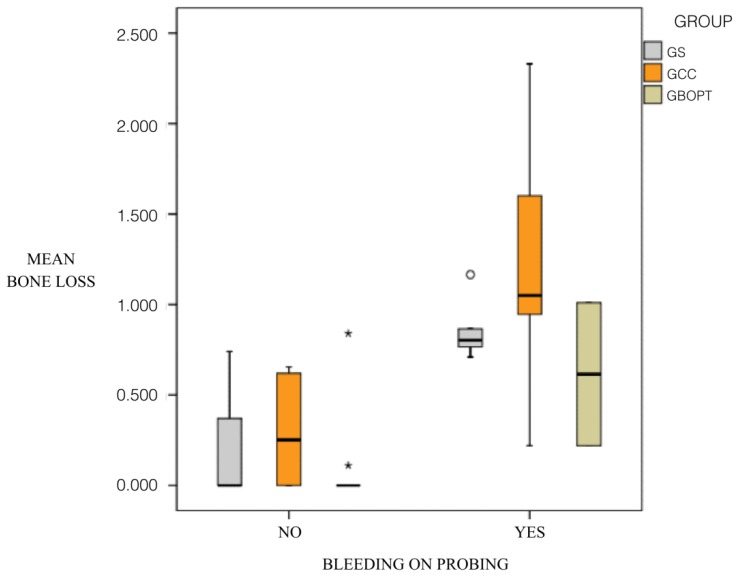
Correlation between the presence of bleeding on probing and bone loss in each group.

**Table 1 jcm-08-02223-t001:** Inclusion and exclusion criteria.

Inclusion Criteria	Exclusion Criteria
Patients with single edentulism (premolar/molar)	Patients with partial or total edentulism i.e., more than a single tooth
>18 years	Patients requiring guided bone regeneration in addition to implant surgery
Willing to attend follow-up visits (3 and 6 months, 1 year, 2 years, and 3 years)	Patients with <7 mm available prosthetic height
Patients in good general health (without any factors compromising implant stability such as previous chemotherapy or radiation to the head and/or neck, progressive active periodontitis, and/or immunosupression)	Pregnant or lactating women
Smokers < 10 cigarettes per day	Patients presenting contraindications for implant placement
Periodontally healthy patients with at least 2 mm vestibular KM	Implants adjacent to edentulous area

**Table 2 jcm-08-02223-t002:** Demographic data and clinical characteristics.

	Female	Male	Total
Number of Patients	49	26	75
Number of Implants	49	26	75
Mean Age	-	-	42.7 ± 10.6

**Table 3 jcm-08-02223-t003:** Percentage of complications (NO/YES) in each prosthetic group.

	Total	Group GS	Group GCC	Group GBOPT
Total	100.0%	100.0%	100.0%	100.0%
No	85.3%	76.0%	81.8%	100.0%
Yes	14.7%	24.0%	18.2%	0.0%

**Table 4 jcm-08-02223-t004:** Relation between clinical peri-implant soft tissue variables and bone loss in the entire sample and the three prosthetic groups.

	Total	Group GS	Group GCC	Group GBOPT
Keratinized mucosa (KMW)	r = −0.88; *p* < 0.001 ***	r = −0.91; *p* < 0.001 ***	r = −0.83; *p* < 0.001 ***	r = −0.68; *p* = 0.001 **
Probing depth (PD)	r = 0.77; *p* < 0.001 ***	r = 0.83; *p* < 0.001 ***	r = 0.85; *p* < 0.001 ***	r = 0.56; *p* = 0.008 **
Bleeding on probing (BOP)	<0.001 ***	<0.001 ***	<0.001 ***	0.019 *
Complications	0.360	0.246	0.053	---

* *p* < 0.05; ** *p* < 0.01; *** *p* < 0.001.

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
