# Peer review of "Influence of Biologically Oriented Preparation Technique on Peri-Implant Tissues; Prospective Randomized Clinical Trial with Three-Year Follow-Up. Part II: Soft Tissues"

_jcm, 2019, doi:10.3390/jcm8122223_

Round 1

Reviewer 1 Report

In regards to the manuscript entitled "Influence of biologically oriented preparation 2 technique on peri-implant tissues; prospective 3 randomized clinical trial with three-year follow-up. 4 Part II: soft tissues". Identified with reference jcm-651665 which I received for review.

The authors presented the results after three years follow-up of two different dental implant restorations. The study is relevant because address the maintenance of the health of the peri-implant soft tissues. Which are the main barrier against bacteria penetration to the implant-bone interface that surrounds the implant neck.

The manuscript is well written, the introduction is appropriate, the materials and methods are clearly described, the results are presented with graphics and in tables, the statistical method is sound, the discussion is developed with relevant and updated literature, and the conclusions are adequately supported by the results. 

There are some comments that require clarification before publication.

Comments to the abstract:

Please replace the following paragraph "The clinical behavior of each restoration type was analyzed after 3 years functional loading by analyzing the surrounding peri-implant soft tissue behavior (keratinized mucosa, probing depth, presence of bleeding) and possible complications." and replace by "After 3 years in function, clinical parameters (presence of keratinized mucosa, probing depths, bleeding on probing, and radiographic bone loss) were compared for the three experimental groups."

Please include a brief summary of the statistical methods in the abstract

Please include in the abstract more information about the results of the study. For example "The cemented restoration without finishing line (BOPT) presented fewer complications and better peri-implant health outcomes in comparison with the other groups. ADD SOME RESULTS DATA THAT SUPPORTS THE SENTENCE"

Comments to the introduction:

Please edit the following sentence "Long term success is partly determined by the peri-implant collar established around the implant." and replace by "Long term implant success is partly determined by the soft tissue collar established around the implant restoration and implant neck because provides an effective seal that protects against bacterial invasion" and add references.

Please edit the following sentence "In addition to radiography (dealt with in Part I of the present work), there are various tools and indicators.." and replace by "In addition to radiography there are various tools and indicators.."

Please insert in the introduction section a paragraph that defines the BOPT preparation, some references about it in teeth, and what benefits could pose in implants to reinforce the justification of the present work. Because there are two parts and the readers may don't have read the first part it is also important to do a brief description.

Comments to Materials and Methods:

In relation to the following sentence "The study sample consisted of 75 implant-supported crowns. Patients were treated between 75 January 9th and December 16th 2015"  During January 9th and December 16th of 2015, what was done? Implant surgery and restorations? The follow-up period started and ended when? Please clarify.

In table 1. In the inclusion criteria side, "Patients with good general health (without any factors compromising implant stability)". Do you mean without additional risk factors?

Comments to Results : NONE

Comments to Discussion: NONE

Comments to Conclusions: NONE

Author Response

Manuscript ID: jcm-651665

Title:
 Influence of biologically oriented preparation technique on 
peri-implant tissues; prospective randomized clinical trial with three-year 
follow-up. Part II: soft tissues.

-Reviewer 1 – 

Comments and Suggestions for Authors

In regards to the manuscript entitled "Influence of biologically oriented preparation 2 technique on peri-implant tissues; prospective 3 randomized clinical trial with three-year follow-up. 4 Part II: soft tissues". Identified with reference jcm-651665 which I received for review.

The authors presented the results after three years follow-up of two different dental implant restorations. The study is relevant because address the maintenance of the health of the peri-implant soft tissues. Which are the main barrier against bacteria penetration to the implant-bone interface that surrounds the implant neck.

The manuscript is well written, the introduction is appropriate, the materials and methods are clearly described, the results are presented with graphics and in tables, the statistical method is sound, the discussion is developed with relevant and updated literature, and the conclusions are adequately supported by the results. 

There are some comments that require clarification before publication.

Reply: Thank you for your comment the article has been modified accordingly.

 Comments to the abstract:

Please replace the following paragraph "The clinical behavior of each restoration type was analyzed after 3 years functional loading by analyzing the surrounding peri-implant soft tissue behavior (keratinized mucosa, probing depth, presence of bleeding) and possible complications." and replace by "After 3 years in function, clinical parameters (presence of keratinized mucosa, probing depths, bleeding on probing, and radiographic bone loss) were compared for the three experimental groups."

Reply: Thank you for your comment. The text has been revised introducing the paragraph as you advise.

Please include a brief summary of the statistical methods in the abstract Please include in the abstract more information about the results of the study. For example "The cemented restoration without finishing line (BOPT) presented fewer complications and better peri-implant health outcomes in comparison with the other ADD SOME RESULTS DATA THAT SUPPORTS THE SENTENCE"

Reply: Thank you. We have added more information the results and statistical methods in the abstract as follows:

Results: Statistical analysis found significant differences in clinical parameters between the different types of crown, the cemented restoration without finishing line (BOPT) presenting fewer complications and better peri-implant health outcomes as: significantly different KMW data (mm), with significant differences between Groups GBOPT and GCC (P<0.001, Kruskal-Wallis test), GBOPT obtaining greater quantities of KM, statistically significant differences between PD values were observed with significant differences between Groups GBOPT and GCC (P=0.010, Kruskal-Wallis test) as well as differences were clearly significant on the presence of BOP between Groups GBOPT and GCC (P=0.018, Chi2 test).  

Comments to the introduction:

Please edit the following sentence "Long term success is partly determined by the peri-implant collar established around the implant." and replace by "Long term implant success is partly determined by the soft tissue collar established around the implant restoration and implant neck because provides an effective seal that protects against bacterial invasion" and add references.

Reply: Thank you for your comment. The text has been revised introducing the sentence as you advise.

Please edit the following sentence "In addition to radiography (dealt with in Part I of the present work), there are various tools and indicators.." and replace by "In addition to radiography there are various tools and indicators.."

Reply: Thank you. We have replaced the sentence as you advise.

Please insert in the introduction section a paragraph that defines the BOPT preparation, some references about it in teeth, and what benefits could pose in implants to reinforce the justification of the present work. Because there are two parts and the readers may don't have read the first part it is also important to do a brief description.

Reply: We have added a description that defines BOPT preparation, and what benefits could pose in implants as follows:

The biologically oriented preparation technique (BOPT) is a type of preparation without finish line. Whit this protocol, the crown’s anatomic emergence profile is eliminated to create a new anatomic crown with a prosthetic emergence profile that simulates the shape of natural tooth [1].

Long term implant success is partly determined by the soft tissue collar established around the implant restoration and implant neck because provides an effective seal that protects against bacterial invasion. This peri-implant soft-tissue union with the implant’s coronal section provides a seal that protects and impedes bacterial invasion and possible future inflammation [10].  This type of restoration without finishing line is also used for implant-supported full coverage crowns, and is known as biologically oriented preparation technique (BOPT). The design improves peri-implant tissue behavior as it eliminates the gap between the restoration and the finishing line at the end of the transepithelial abutment leaving the apical portion of the abutment free of coverage by the prosthetic restoration for at least 2 mm in order to stabilize the adjacent connective tissue. It has also been suggested that a minimum peri-implant mucosa thickness is needed to protect osteointegration after placing the abutment and loading the implant [9, 11].

Comments to Materials and Methods:

In relation to the following sentence "The study sample consisted of 75 implant-supported crowns. Patients were treated between 75 January 9th and December 16th 2015"  During January 9th and December 16th of 2015, what was done? Implant surgery and restorations? The follow-up period started and ended when? Please clarify.

Reply: Thank you for your comment. The text has been revised introducing fthe explanation  as you advise:

The study sample consisted of 75 implant-supported crowns. Patients were treated between January 9th and December 16th 2015. Implant surgeries as well as the placement of the prosthetic restoration were performed during this period, and starting the follow-up period in January 2016 and finishing in January 2019.

In table 1. In the inclusion criteria side, "Patients with good general health (without any factors compromising implant stability)". Do you mean without additional risk factors?

Reply: Thank you. We have added a short description of the risk factors.

Patients in good general health (without any factors compromising implant stability such as previous chemotherapy or radiation to the head and/or neck, progressive active periodontitis and/or immunosupression)  

Comments to Results : NONE

Comments to Discussion: NONE

Comments to Conclusions: NONE

Reviewer 2 Report

Dear authors,

thank you for the submission of your present work! 

You investigated the influence of 3 preparation techniques on peri-implant tissues and conducted a PRCT, which is appropriate.

However, there are some main issues that have to be resolved before this manuscript can be considered for publication. 

The main issue is that it has to be made clear in every part of the publication that it is a unique work! You refer repeatedly to another work that has even not be accepted yet and you cite the results of the other work. Moreover you refer to the results of the hard tissue paper and do statistical analyses although the aim of this work, according to the title, should be the soft tissue. This is not acceptable! Or do you say that THIS article is not a unique work? Then to you would have to sum up both articles in one!

Furthermore, in the results section, you did not give information on the KMW situation before implant placement in the groups in detail (you must have measured it during implant placement after crestal incision?). How can you give the KMW then as a result and relate it to the different prosthetic types? Please give information on that.

The work may be acceptable after thorough major revision. Please look at the comments that I made.

Kind regards

Author Response

Manuscript ID: jcm-651665

Title:  Influence of biologically oriented preparation technique on 
peri-implant tissues; prospective randomized clinical trial with three-year 
follow-up. Part II: soft tissues.

-Reviewer 2 – 

Comments and Suggestions for Authors

Dear authors,

thank you for the submission of your present work!

You investigated the influence of 3 preparation techniques on peri-implant tissues and conducted a PRCT, which is appropriate.

However, there are some main issues that have to be resolved before this manuscript can be considered for publication.

Reply: Thank you for your comment the article has been modified accordingly.

The main issue is that it has to be made clear in every part of the publication that it is a unique work! You refer repeatedly to another work that has even not be accepted yet and you cite the results of the other work. Moreover you refer to the results of the hard tissue paper and do statistical analyses although the aim of this work, according to the title, should be the soft tissue. This is not acceptable! Or do you say that THIS article is not a unique work? Then to you would have to sum up both articles in one!

Reply: Thank you for your comment. The text has been revised introducing the explanations as you advise. We have divided the investigation in two parts: part I about hard tissues and part II about soft tissues. That is why we refer to the results of hard tissue paper. We have still explained that measurements of radiograph  bone loss  were made in order to correlate the loss either with the main variables (soft tissue variables) of this investigation as follows:

2.2. Bone loss (mm) analysis to allow the subsequent correlation

To allow the correlation between soft tissues behaviour and the possible influence of  bone loss, peri-implant bone changes were evaluated using standardized parallelized periapical radiographs. As the implants were placed at crestal bone level, the straight line between the mesial and distal sides of the implant was considered as height 0 (corresponding to the implant platform). Measurements were taken to evaluate bone changes between T0 (prosthetic loading) and T1 (3 years later). To determine bone loss, a line was traced perpendicularly from points on the mesial and distal aspects of each implant to the most coronal bone level. In this way, bone loss was measured on both mesial and distal sides of the implant on each radiograph. The measurements were taken using 3D modeling software (Rhinoceros®, Robert McNeel & Associates, Seattle, U.S.A.).

All the variables analyzed in Part I and Part II of  in the present trial were correlated to determine any significant relations between them.

Bone loss (mm) for the three types of prosthesis was also analyzed using radiographs, as explained in the material and method section, to allow the correlation with the variables recorded. Radiographic peri-implant bone loss showed differences between the three types of restoration used: Group GS (screw-retained) = 0.35 ± 0.37mm; Group GBOPT (cemented BOPT) = 0.10 ± 0.28mm; and Group GCC (conventional cementation with finishing line) = 0.67 ± 0.62 mm. Median bone loss values were: 0.36; 0.63; and 0.00 mm, respectively.

 The relation between clinical peri-implant soft tissue variables and bone loss is shown in the following table (Spearman’s coefficient and Mann-Whitney test) (Table 5).

Furthermore, in the results section, you did not give information on the KMW situation before implant placement in the groups in detail (you must have measured it during implant placement after crestal incision?). How can you give the KMW then as a result and relate it to the different prosthetic types? Please give information on that.

Reply: Thank you for your comment. The text has been revised introducing the explanations as you advise. We have measured at the time of the prosthetic load to have a crown as a reference, and then, 3 years after loading.

The clinical parameters recorded included the width of keratinized mucosa (KMW) measured in the vestibular area of peri-implant mucosa. KMW was measured from the mucosa margin adjacent to  of the crown to the mucogingival junction (or even free mucosa) placing the probe parallel to the longitudinal axis of the implant as proposed in a study by Schwarz et al. [26], the first measurement was made at the time of the prosthetic load to have the crown as a reference of an immovable zone on different measurements (3 years after loading). PD was recorded, as well as BOP, as described by Ainamo and Bay [27].

Round 2

Reviewer 2 Report

Dear authors,

thank you for implementation of my recommendations.

The quality of your paper has significantly increased, but as long as the results of the bone loss are part of your other paper (and will be published), I would still recommend to remove these results from this paper.

Kind regards

Author Response

Thank you for taking the time to evaluate our manuscript and for your advice! Please see the attachment.
